# Challenges of Managing Type 3c Diabetes in the Context of Pancreatic Resection, Cancer and Trauma

**DOI:** 10.3390/jcm13102993

**Published:** 2024-05-19

**Authors:** Colton D. Wayne, Chahrazed Benbetka, Gail E. Besner, Siddharth Narayanan

**Affiliations:** 1Department of Pediatric Surgery, Nationwide Children’s Hospital, 700 Children’s Drive, Columbus, OH 43205, USA; colton.wayne@nationwidechildrens.org (C.D.W.); gail.besner@nationwidechildrens.org (G.E.B.); 2Center for Perinatal Research, Nationwide Children’s Hospital, Columbus, OH 43205, USA; 3Department of Surgery, Baylor University Medical Center, 3600 Gaston Ave, Dallas, TX 75246, USA; 4Faculty of Pharmacy, University of Algiers 1, Algiers 16000, Algeria; chahrazed.benbetka1@gmail.com

**Keywords:** type 3c diabetes mellitus, pancreatic resection, pancreatic cancer, pancreas, insulin, partial pancreatectomy, total pancreatectomy, trauma

## Abstract

Type 3c diabetes mellitus (T3cDM), also known as pancreatogenic or pancreoprivic diabetes, is a specific type of DM that often develops as a result of diseases affecting the exocrine pancreas, exhibiting an array of hormonal and metabolic characteristics. Several pancreatic exocrine diseases and surgical procedures may cause T3cDM. Diagnosing T3cDM remains difficult as the disease characteristics frequently overlap with clinical presentations of type 1 DM (T1DM) or type 2 DM (T2DM). Managing T3cDM is likewise challenging due to numerous confounding metabolic dysfunctions, including pancreatic endocrine and exocrine insufficiencies and poor nutritional status. Treatment of pancreatic exocrine insufficiency is of paramount importance when managing patients with T3cDM. This review aims to consolidate the latest information on surgical etiologies of T3cDM, focusing on partial pancreatic resections, total pancreatectomy, pancreatic cancer and trauma.

## 1. Introduction

Type 3c diabetes mellitus (T3cDM) is a distinct type of DM that occurs in the context of diseases of the exocrine pancreas. It is also known as pancreoprivic, pancreatogenic, or secondary DM (secondary to diseases that affect the pancreas exocrine compartment) [1]. The most common etiologies for T3cDM include acute pancreatitis (AP), acute recurring pancreatitis (ARP), chronic pancreatitis (CP), cystic fibrosis (CF), as well as understudied causes such as pancreatic resections, cancer, or genetic disorders [2,3].

A widely accepted paradigm related to pancreatitis is the concept of a disease continuum, where 30% of patients with AP tend to develop a form of chronic disease, either with ARP or CP [4]. A single attack of pancreatitis can result in T3cDM, but the risk is highest with chronic or recurrent episodes [1]. AP is categorized into mild, moderate, and severe forms, with severe AP leading to a greater incidence of new onset DM than mild AP [5]. However, there is a disparity regarding the extent to which AP severity affects T3cDM development [6]. On the other hand, patients diagnosed with CP have a 70% likelihood of developing secondary DM, with its incidence increasing to 90% in patients with calcific CP [7].

While previously considered rare, it is now known that the actual incidence of T3cDM is much higher within the general patient cohort with DM [8], affecting 5–10% of all patients with DM. However, accurate estimates are difficult to obtain as the disease is often misdiagnosed [2]. A recent report has shown that the incidence of T3cDM is much higher in children with ARP and CP (4–9%) than the overall frequency of DM (0.25%) in the general pediatric population [9]. T3cDM risk may be elevated in children as age progresses or with other associated comorbidities such as pancreatic exocrine insufficiency, obesity, or pancreatic atrophy [10,11]. Total pancreatectomy with islet autotransplantation (TPIAT) is a viable treatment option considered for a subgroup of patients with ARP, CP or severe forms of hereditary pancreatitis [12] when all other treatments have failed to manage the disease. Because of the total pancreas resection and islet loss, TPIAT is associated with a high rate of post-surgical insulin dependent DM [13]. Consequently, this select group of children undergoing TPIAT for intractable ARP or CP are not included when determining the 4–9% prevalence of pancreatogenic DM in children [9].

T3cDM is a complex and multifactorial disease that is often misdiagnosed as either T1DM or T2DM [14]. Table 1 compares some of the known characteristics of T3cDM with those of T1DM and T2DM. The initial diagnosis for T3cDM is like that of T1DM and T2DM but differentiating it from the latter two is more complicated [7,15]. While T1DM is autoimmune-mediated and beta cell specific [6], T3cDM clinical presentation can be heterogeneous, exhibiting several conditions marked by varying levels of endocrine and exocrine impairments [7]. Likewise, studies indicate that patients with T3cDM have a reduced body mass index and are devoid of features linked to the metabolic syndrome associated with T2DM [16]. According to a study that included age- and sex-matched cohorts, individuals with post-pancreatitis DM have a higher risk of hospitalization and mortality compared to those with T2DM [17]. Additionally, unlike T1DM and T2DM, patients with T3cDM experience mild glucose intolerance that often progresses to brittle diabetes that is challenging to manage, marked by fluctuating hyper- and hypo-glycemic swings [3].

Although diagnostic criteria have been proposed for T3cDM [7], they are not standardized and pose difficulty to implement in various clinical settings [24]. These diagnostic criteria are available and useful at present but are not absolute due to a potential overlap between different types of DM. For example, it is known that pancreatic exocrine insufficiency and pancreatic atrophy are found in long-standing T1DM or T2DM. Additional factors that make it extremely challenging to differentiate T3cDM from T1DM or T2DM include factors such as: (i) the occurrence of T1 or T2DM independent of exocrine dysfunction in patients with AP or CP [25], (ii) the incidence of DM attributing to a high risk of developing AP, CP, or pancreatic cancer [2,7], or (iii) the impact of decreased physician awareness for a T3cDM misdiagnosis [7].

T3cDM can also develop in the setting of surgical intervention on the pancreas and includes procedures such as partial pancreatectomy (PP) and total pancreatectomy (TP) for several indications including cancer (Table 1). In this review, we discuss the impact of underexamined etiologies, including pancreatic surgical procedures, pancreatic cancer, and pancreatic trauma, on the development of T3cDM, and the challenges in managing it (Figure 1).

## 2. Pancreatectomy

### 2.1. Partial Pancreatic Resections

PP is performed for several pathologies, including malignancy, CP, and trauma. In the context of CP, some surgical treatments include the Puestow procedure for drainage, the Beger or the Kausch-Whipple procedure for resection, and the Frey procedure combining drainage and resection [26]. The choice of the procedure is determined based on the underlying cause of the pancreatitis and the relevant anatomy of the patient [26,27]. Despite the efficiency of these procedures in reducing the debilitating pain associated with the disease, they often result in relapse [22]. Another limitation is the high risk of developing T3cDM after partial pancreatic resection, which is considered the most feared complication [18]. Both exocrine and endocrine dysfunction after PP have been documented with rates as high as 35% and 40%, respectively [24]. Pre-existing hyperglycemia, elevated HbA1c, older age, female sex, and obesity are recognized as preoperative factors that impact the development of T3cDM after PP [19]. The portion of the pancreas resected, and the extent of pancreatic resection performed also play significant roles in the severity of the T3cDM that occurs [18].

There are different procedures for PP, including the Whipple procedure (pancreaticoduodenectomy; PD), duodenal-sparing pancreatic head resection, central pancreatectomy and distal pancreatectomy (DP) (Figure 2). Patients undergoing DP have a higher incidence of postoperative T3cDM compared to those undergoing other resections. This is because a significant proportion of the pancreatic islet cells and insulin-producing beta cells exist in the tail region of the pancreas [20,21]. Kang et al. found that 50% of patients who underwent DP developed glucose intolerance one year after surgery [19]. Interestingly, Burkhart et al. observed that DP was associated with increased incidence of new onset postoperative T3cDM compared to the pancreaticoduodenectomy (PD; Whipple procedure). However, patients who underwent the Whipple procedure were more susceptible to worsening of pre-existing DM than those who underwent DP [28]. Overall, it is important to closely monitor patients who require DP for the development of T3cDM (Table 2).

Central pancreatectomy is not common but may be performed for tumors in the body or neck of the pancreas that are too large for simple enucleation [29,30,31,32]. During this procedure, preserving the body and tail of the pancreas may result in a lower incidence of T3cDM, but it also carries a higher risk of complications such as pancreatic fistulas and leaks due to the two residual portions of pancreatic tissue that remain after surgery [33,34,35]. When compared to PD twelve months after surgery, there was no significant difference in overall morbidity, exocrine dysfunction, postoperative fistula formation, or remnant pancreatic volume [36]. Unless significant pancreatic tail parenchyma can be saved, central pancreatectomy does not appear to improve overall outcomes compared to other extensive resections.

Other surgical procedures have been developed to treat pathologies in the body and head of the pancreas. The more extensive methods are the Whipple procedure and the duodenal-sparing pancreatic head resection, which are commonly utilized when tumors are present [18,37]. However, these procedures have a postoperative incidence of 9–24% of T3cDM in patients [18,38]. Drainage procedures also exist, allowing for targeted excision of certain areas of the pancreas while incising the pancreatic duct longitudinally and performing an enteric anastomosis to enable pancreatic remnant drainage [39]. These include the modified Puestow (longitudinal pancreatojejunostomy (PJ)), the Beger (resection of the pancreatic head with end-to-side PJ), and the Frey (coring out of the pancreatic head with longitudinal PJ) procedures, among others, and may be utilized in the setting of malignancy or surgical intervention for unremitting pain in CP [20,40]. It is essential to mention conflicting data in the literature regarding whether T3cDM consistently develops after these procedures, but this has been reported in smaller studies [41,42,43].

### 2.2. Total Pancreatectomy (TP)

TP has been extensively documented as a surgical procedure used to treat CP and pancreatic cancers [26,44,45,46,47]. The procedure is associated with significant morbidity, requiring advanced care and meticulous management to handle post-surgery metabolic derangement [48]. TP involves the excision of the entire pancreas, resulting in the loss of both exocrine and endocrine function (Table 2). This leads to absolute insulinopenia, requiring a lifelong need for exogenous insulin therapy. Studies evaluating T3cDM after TP have shown that these patients suffer from brittle diabetes characterized by drastic hypo- and hyperglycemic swings [2,3,49]. A recent multicenter study from Japan demonstrated that patients who had undergone TP mainly for pancreatic cancers had inadequate glycemic control within the first postoperative year and experienced frequent hypoglycemic events [50].

After TP, patients experience a more severe T3cDM disease than those with residual functioning endocrine and exocrine pancreas after PP [18]. This is because TP patients suffer from deficiencies in insulin, glucagon, amylin, and pancreatic polypeptide, and as a result, they will often develop hyperglycemia due to increased hepatic gluconeogenesis leading to decreased hepatic insulin sensitivity [51]. Conversely, post-pancreatectomy hypoglycemic episodes occur after insulin administration, when normal or enhanced peripheral insulin sensitivity causes a significant drop in systemic glucose levels that can no longer be countered by pancreatic glucagon production [8]. A recent report suggests that developing new-onset T3cDM after TP has unique pathophysiological and clinical features, including endocrine abnormalities, a deficit of pancreatic enzymes, and TP-related alterations of gastrointestinal anatomy, therefore requiring a new nomenclature [22,52]. Interestingly, some studies support that post-TP DM may not be as severe as previously thought [53,54], but further research is needed to address glycemic outcomes and overall quality of life (QoL) following TP.

Pancreatic enzyme loss is a major factor in the development of T3cDM after TP surgery [46,55]. Even a partial loss in pancreatic exocrine function can affect the absorption of nutrients, including fat soluble vitamins, particularly vitamin D [7]. After TP, patients must rely entirely on the exogenous administration of pancreatic enzymes. Therefore, it is important to carefully monitor pancreatic enzyme replacement therapy (PERT) in this population to prevent the progression of T3cDM severity, which is closely linked to loss of exocrine pancreatic function [55].

Total pancreatectomy with islet autotransplantation (TPIAT) is an established surgical therapy for a subgroup of adult and pediatric patients with CP and ARP (Figure 2). The procedure aims to improve QoL and prevent or reduce the severity of post-TP DM and hypoglycemic awareness [47,56,57,58]. Although the TP procedure can alleviate debilitating pain associated with CP, it can be potentially devastating as it causes obligatory DM, which is why it is combined with IAT. The dual procedure helps preserve endogenous beta cell mass and insulin secretory capacity to mitigate post-TP-induced DM as much as possible (Table 2). For the IAT therapy, the remaining functional islet cells from a diseased CP pancreas are salvaged by ductal infusion of pancreatic tissue dissociation enzymes to obtain an islet equivalent volume. The harvested islets are then reimplanted into the host liver, the preferred transplantation site. Over time, this procedure has shown promising outcomes for glycemic control post-transplantation [59,60]. These patients do require PERT postoperatively to combat the loss of pancreatic exocrine function [60].

Several studies have shown the incredible potential of TPIAT as a viable approach to preventing and treating post-pancreatectomy DM. One of the earliest and largest studies evaluated the long-term outcomes of TPIAT ten years after the surgery in adults and pediatric patients (*n* = 215; 185 adults and 30 children) and showed that it resulted in excellent pain relief and sustained islet graft function [61]. The study found that pediatric patients were more likely to have islet function than adults, and that the infusion of islet equivalents per kilogram bodyweight of greater than 4,000 was the strongest predictor of islet graft function [61]. Another recent report on long-term outcome and durability over ten years (*n* = 142; age range, 14–62 years) demonstrated exceptional long-term pain control, robust improvements in QoL, a decline in islet cell function, but overall stable maintenance of glycemic control [52]. On the other hand, some studies have pointed out the limitations of the TPIAT approach. For example, the latest report from the POST consortium, which evaluated nutritional risks in 348 patients undergoing TPIAT, showed that the prevalence of fat-soluble vitamin deficiencies increased after the procedure, especially in underweight patients [21]. Therefore, experts from this panel strongly recommended close postoperative monitoring for nutrient and vitamin levels. Additionally, the POST consortium conducted the first multicentric prospective study involving 11 centers within the USA, comparing children (*n* = 84) with adults (*n* = 195) undergoing TPIAT to evaluate surgical approaches and the early postoperative course [47]. This study identified important differences in pancreatectomy techniques, islet isolation approaches, outcomes, and complications. Prior pancreatic surgery has a significant impact on TPIAT outcomes [21,47,59,62,63].

Despite its limitations, recent studies have shown that metabolic testing performed three months after TPIAT strongly correlated with later diabetes outcomes, offering a reliable prediction model that provides valuable prognostic insight early after TPIAT [64]. Furthermore, modified management protocols, including early use of insulin pumps post TPIAT in pediatric patients, have demonstrated enduring benefits [65,66].

## 3. Pancreatic Cancer

In addition to AP and CP, T3cDM can occur in the setting of pancreatic malignancy [63,67], with data from a large single-center review suggesting that pancreatic cancer may be the cause of about 8% of all T3cDM cases [67]. This is particularly relevant in the context of pancreatic ductal adenocarcinoma (PDAC), which has a direct pathologic relationship to the inflammation caused by CP or RAP [68,69].

It is estimated that PDAC will be the second leading cause of cancer death in the USA by 2030 [67]. Patients often present with symptoms of disseminated disease, such as jaundice, weight loss, fatigue, and pale stools, before their primary pancreatic tumor is identified [70,71]. Moreover, patients with new-onset DM may be diagnosed with T3cDM and an occult pancreatic malignancy; this is why many studies have suggested that this is an indication for pancreatic cancer screening in these unique scenarios [68,72,73,74]. Although the correlation between CP, diabetes, and pancreatic malignancy is well-known, clinicians tend to screen for pancreatic cancer in patients with CP more often than those with hyperglycemia and a new-onset diabetes [73].

On the other hand, studies have shown a strong association between diabetes (T1DM, T2DM, and T3cDM) and pancreatic cancer. Patients with DM have a higher relative risk of pancreatic cancer [72,74,75,76]. In fact, Butler et al., using autopsy and pathology specimens, found that patients with obesity, DM, and CP all had increased replication of pancreatic ductal epithelium and a higher likelihood of developing pancreatic cancer [77]. While typical demographic features and family history do not facilitate distinguishing T2DM from DM secondary to pancreatic cancer, weight loss at the time of DM onset is more prevalent in PDAC than in T2DM [78]. Furthermore, DM has been associated with negative clinical outcomes in patients having PDAC and can be a predictor of worse survival at all stages of the disease [67]. Patients with DM also have an increased risk of complications after surgery for PDAC [79]. Recent studies have suggested potential biomarkers such as pancreatic polypeptide, adiponectin, and IL-Ra to distinguish T3cDM from T2DM, which may likely facilitate earlier detection of pancreatic cancers in this patient population [80,81,82]. Nevertheless, more research is needed to determine the frequency of occult pancreatic cancers present in patients with new-onset DM.

Inflammation in the pancreatic parenchyma can destroy islet cells and result in impaired endocrine function and the development of pancreatogenic diabetes [67,83]. This can occur anywhere along the spectrum of disease progression from CP to PDAC [68,83,84]. Humoral aberrations and paraneoplastic phenomena, such as impaired glucose metabolism in skeletal muscle and insulin secretory capacity, have also been implicated in the development of pancreatogenic diabetes resulting from PDAC [24,85], and a recent study suggesting that PDAC-associated T3cDM is clinically distinguishable compared to patients having conventional T2DM [24]. While most research on T3cDM and pancreatic malignancy focus on PDAC, which accounts for 90% of all pancreatic cancers [86], fewer studies have been done in other pancreatic lesions, such as pancreatic neuroendocrine tumors (pNETs), intraductal papillary mucinous neoplasms (IPMNs), and mucinous cystic neoplasms. Firkins et al. recently showed that 20% of 311 patients without previous pancreatic disease developed new-onset DM within two years following PP for pancreatic cystic lesions [87]. The risk of developing new-onset DM was reported to be similar following DP vs. PD for cystic lesions (88). Other meta-analyses have reported that the risk of developing new-onset DM was 16% following PD and 14% following DP, where pancreatectomy (after excluding CP) was primarily limited to benign or malignant lesions [20,88].

Non-functional pNETs are the most common type of pNET associated with T3cDM, especially after tumor resection or enucleation [89]. On the other hand, functional pNETs such as insulinoma, glucagonoma, and somatostatinomas can lead to DM, but they are classified in a different subgroup of T3cDM known as T3dDM [89,90]. IPMNs and mucinous cystic neoplasms are less common and account for only 1% of all pancreatic carcinomas [91,92]. While the destruction or dysfunction of islet cells may occur during the late stages of these tumors, the mass effect of large tumors and ductal obstruction may also contribute to the development of endocrine dysfunction, leading to T3cDM [93,94]. As with PDAC, new-onset hyperglycemia may indicate the development of T3cDM in patients with undiagnosed pNETs or cystic neoplasms and should immediately prompt clinicians to initiate proper screening protocols.

Over the years, the survival rates for pancreatic cancer patients have increased due to the combination of advanced imaging technologies, improved early detection methods, and better postoperative care [18,95,96]. Consequently, more patients diagnosed with pancreatic cancer undergo operative resection, leading to a higher incidence of developing T3cDM postoperatively [97]. A study conducted by Wu et al. over ten years showed that the median time for T3cDM to develop and be diagnosed after a TP procedure was 12–15 months [37]. Although it is an incredibly positive trend that more patients are surviving operative treatment for pancreatic cancer, it is imperative to realize the risk of T3cDM in these patients and to address this quickly due to its long-term morbidity and mortality implications.

## 4. Pancreatic Trauma

In addition to the etiologies mentioned earlier, traumatic pancreas injury can also contribute to the development of T3cDM. The most common cause of pancreas injury is blunt trauma due to motor vehicle collisions that cause acceleration-deceleration and compression injuries, while penetrating injury accounts for a smaller percentage of pancreas trauma [98,99]. Injuries can be graded into different levels to establish severity: Grade 1 and 2 are less severe contusions or lacerations that spare the duct, grade 3 injuries involve the duct at the body or tail, grade 4 injuries involve the duct at the head, and grade 5 injuries result in severe disruption of the pancreatic head [100]. The severity of the injury determines whether operative resection is necessary [101].

In a study of pancreatic trauma patients conducted over 30 years at a single center, it was found that 35.7% of patients developed diabetes post-discharge. Additionally, 50% of patients who underwent some degree of pancreas resection developed diabetes after surgery [102]. Another small study revealed that 15.8% of pancreatic trauma patients had some extent of endocrine dysfunction between 1 month and 4 years after injury [103]. In contrast, some studies haven’t reported any endocrine impacts prior to discharge in patients requiring PP for pancreatic trauma. It is important to note that these studies did not conduct any follow-up after discharge and may have missed the delayed development of T3cDM after trauma [104,105,106]. Although a significant number of patients who developed T3cDM after pancreatic trauma had undergone operative intervention, there are reports of patients with non-operative pancreatic injuries that develop diabetes and endocrine dysfunction [102,107]. Therefore, clinicians need to screen for T3cDM development and progression in all patients who have experienced pancreatic trauma, regardless of whether or not operative resection was performed.

## 5. Management

Patients with T3cDM have similar risks for micro- and macrovascular complications as those observed in T1DM and T2DM [3]. Despite efforts to understand T3cDM pathophysiology, much of its current management relies heavily on data from studies focused on treating T1DM, T2DM, or AP, CP, and CF. From a clinical standpoint, the associated pancreatic disorders are diverse, with early evidence supporting the concept that mechanisms of hyperglycemia differ in various forms of T3cDM [67].

While early and periodic screening for DM with CP is recommended, the same is also critical for non-diabetic patients with CP as they are strongly predisposed to T3cDM [6]. Patients diagnosed with CP without DM exhibit an extended glucose tolerance curve, and once diagnosed with CP, 40% of patients exhibit early-onset DM [108]. Another study also showed that a two-hour oral glucose tolerance test helped in early detection of glycemic alteration even when fasting blood glucose and HbA1c were normal, indicating that an appropriate management could help prevent CP severity [109]. Furthermore, from a molecular standpoint, Bach2 gene has been identified as a potential biomarker contributing to the immune mediated chronicity in CP [110], but how this biomarker links to management of T3cDM in CP and other etiologies remains to be evaluated. Children diagnosed with ARP and CP are at high risk of developing T3cDM. Thus annual screening for diabetes, including fasting glucose and HbA1c levels, has been recommended for them [9]. It is important to mention that there are limited data related to the pathophysiology of T3cDM in children, as nearly all mechanistic studies have been conducted in adults [9]. Continuous glucose monitoring has been supported in recent years as the best way to monitor glucose levels in patients with brittle diabetes and potentially profound glycemic swings [1].

Management of T3cDM, including dietary management, is still an area of active debate among clinicians. It requires a multidisciplinary team that addresses and offers guidance on other critical comorbidities, such as pancreatic exocrine insufficiency and nutritional constraints [9]. Despite T3cDM representing a unified category of secondary diabetes, it has been suggested that different treatment options must be considered and tailored individually based on the T3cDM pathophysiological background [2]. Insulin therapy has been the earliest treatment for pancreatogenic diabetes since T3cDM was initially categorized as a specific subtype of DM [8,9]. It has a desirable anabolic effect in malnourished patients, but there is no consensus on the optimal regimen. It is important to appropriately dose insulin to avoid hypoglycemic episodes in patients with T3cDM who may have peripheral insulin hypersensitivity [2,111]. Though insulin therapy remains the standard treatment regimen, clinical trials of diabetes treatment in this unique group are warranted. Recent studies have shown that insulin therapy may lead to an increased risk of PDAC and must be used with caution in T3cDM patients with existing inflammatory pancreas pathology [72,112]. Therefore, patients must be carefully screened prior to initiating insulin therapy.

The HaPanEU study indicated that most patients do not respond satisfactorily to oral glycemic agents and recommend switching to insulin treatment [113]. Some studies have suggested that metformin is the most effective treatment for T3cDM in patients who have undergone PP but still have functioning islet cells [112,114]. In cases of mild hyperglycemia, metformin can help improve the insulin peripheral effect while reducing hepatic glucose production [115]. Though metformin helps to treat insulin resistance, gastrointestinal side-effects must be monitored. Furthermore, research has shown that patients with T3cDM on metformin therapy have a lower overall risk of pancreatic cancer [116,117,118]. On the other hand, glucagon-like peptide 1 (GLP1) analogs may increase the risk of pancreatic cancer and are not recommended for pancreatogenic diabetes [112,119,120]. Multiple studies using mouse and rat models have shown that GLP1 analogs increased *PDX1* expression, leading to the transformation of pancreatic ductal cells to a malignant phenotype [121,122]. There are no new data available on the use of potential sodium-glucose cotransporter 2 inhibitors for patients with T3cDM [2]. Overall, further investigations are needed to provide recommendations on the best clinical practice guidelines for the pharmacological management of T3cDM.

## 6. Conclusions

Data focused on AP, CP, and CF have played a pivotal role in enhancing our understanding of T3cDM pathophysiology, especially those related to under-evaluated etiologies including surgery for CP, cancer, and pancreatic trauma. The development of T3cDM after partial or total pancreatic resections is a severe complication that requires careful monitoring and systematic management. A thorough history and metabolic-endocrine assessments before and after pancreatic surgery are essential aspects of the perioperative care of T3cDM. New onset of T3cDM in patients may be indicative of previously undiagnosed pancreatic cancer. It is important to manage other severe comorbidities, such as exocrine pancreatic insufficiency and nutritional deficiencies, in these patients.

## Figures and Tables

**Figure 1 jcm-13-02993-f001:**
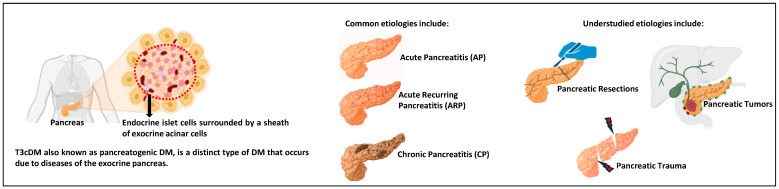
**Graphical abstract.** This review aims to consolidate the latest information on understudied etiologies of T3cDM, focusing on partial pancreatic resections, total pancreatectomy, pancreatic cancer, and trauma.

**Figure 2 jcm-13-02993-f002:**
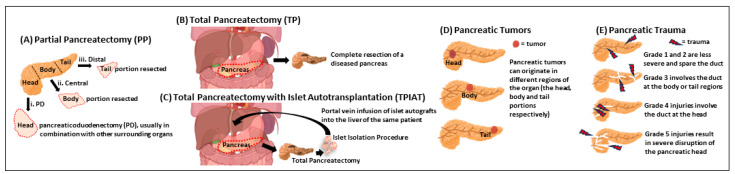
**Surgical methods for pancreatectomy, locations of cancer in the pancreas and grading for pancreatic trauma**. The choice of pancreatic surgery is based on the underlying cause and the relevant pancreatic anatomy. The pancreas can be broadly divided into head, body and tail regions. (**A**) Partial pancreatectomy (PP) involves resecting either: (i) the head region in combination with some of the surrounding organs (e.g., pancreaticoduodenectomy, PD); (ii) the body region (for a central pancreatectomy, which is not commonly performed); or (iii) the tail region for a distal pancreatectomy (DP, sometimes performed in combination with the body region). (**B**) A total pancreatectomy (TP), often used to treat chronic pancreatitis and pancreatic cancers, is a complex surgical procedure where the entire diseased pancreas is excised, and involves removal of other surrounding organs, vessels and lymph nodes. One of the major limitations of this procedure is life-long dependence on exogenous insulin and enzyme replacements. (**C**) T3cDM is a major challenge after a TP procedure. In some cases, a simultaneous islet auto transplantation (IAT) procedure is performed after TP. This dual TPIAT therapy helps to preserve endogenous beta cells and insulin secretion to alleviate post-TP-induced DM. For IAT, the remaining functional islet cells from a diseased organ are recovered by digesting the pancreas with tissue dissociation enzymes to obtain an islet equivalent volume. The harvested islet autografts are then reimplanted into the host liver via portal vein infusion. (**D**) In the context of pancreatic cancers, a tumor can originate in the head, body or tail regions of the pancreas. The development of T3cDM can occur anywhere along the progression of disease spectrum, beginning with an inflammatory insult. Some types of pancreatic cancers include the Pancreatic Ductal Adeno Carcinoma (PDAC), pancreatic NeuroEndocrine Tumors (pNETs), Intraductal Papillary Mucinous Neoplasms (IPMNs) and the Mucinous Cystic Neoplasms. (**E**) T3cDM can also occur after pancreatic trauma, with traumatic injuries graded (1–5) based on the severity of the injury.

**Table 1 jcm-13-02993-t001:** Features of Type 1, Type 2 and Type 3c Diabetes Mellitus.

Factor	T3cDM	T1DM	T2DM
Age of Onset	Any age	Childhood/teens (however, must be carefully diagnosed from other forms of childhood DM)	Typically, adulthood
Overweight/Obese	Uncommon (in some cases, despite being overweight muscle depletion is observed)	Rare	Common (known risk factor)
BMI	Normal-Reduced range	Variable	Usually, high
DM-associated antibodies/autoimmunity	Absent/lacking	Present/present	Rare/absent
Diabetic Ketoacidosis	rare	Common	rare
Hyperglycemia	Mild to severe (in brittle DM)	Severe	Usually, mild
Hypoglycemia	Common (can be severe)	Common	rare
Hepatic Insulin Insensitivity	Decreased	Normal	Normal or decreased
Peripheral Insulin sensitivity	Increased	Normal or increased	decreased
Insulin levels	Low	Low	High
Glucagon levels	Low	Normal or high	Normal or high
Pancreatic Polypeptide levels	Low	Normal or low	High
Malnutrition/Nutrient deficiency	Common/Deficiency of fat-soluble vitamins (associated with EPI + poor diet)	Uncommon/rare	Rare/rare
EPI	Can precede and/or is more pronounced in patients with T3cDM often accompanied with metabolic/nutritional derangements	Can be observed in patients having long-standing T1DM	Can be observed in patients having long-standing T2DM
BMD	Risk of having low BMD substantial depending on type of exocrine pancreatic disease	Risk of having low BMD	May have low BMD
**T3cDM in the context of PP, TP and PDAC:** The portion of resected pancreas and the extent of PP determine the severity of the ensuing T3cDM [18]. Pre-existing hyperglycemia, elevated HbA1c, older age, female sex, and obesity are recognized preoperative factors impacting development of T3cDM after PP [19]. DP results in a higher incidence of postoperative T3cDM compared to other resections [20,21]. T3cDM is a lot more severe and difficult to manage after a TP procedure (‘brittle DM’ is a hallmark feature with frequent hyper- and hypoglycemic swings). The development of new-onset DM after TP has called for a new nomenclature owing to its several unique pathophysiological and clinical features including endocrine abnormalities, a deficit of pancreatic enzymes, and TP-related alterations of GI anatomy [22]. PDAC-associated T3cDM features include older patients, lower BMI, higher ALT levels, and relatively lower HbA1c levels. Patients also exhibit classic PDAC signs: jaundice, GI symptoms, weight loss, greater insulin usage. No single associated risk factor is predictive alone [23].			

Abbreviations: ALT, alanine aminotransferase; BMD, bone mineral density; BMI, body mass index; DM, diabetes mellitus; EPI, exocrine pancreatic insufficiency; GI, gastrointestinal; HbA1c, glycated hemoglobin; PDAC, pancreatic ductal adenocarcinoma; PP, partial pancreatectomy; TP, total pancreatectomy. Information in Table 1 adapted from the following references: [15,18,19,20,21,22,23].

**Table 2 jcm-13-02993-t002:** Summary of the advantages and disadvantages of different types of pancreatic surgery.

Type of Pancreatic Surgery	Advantages	Disadvantages
**PP resections:**	Performed based on the underlying cause of disease and relevant anatomy. Indicated for several pathologies including malignancies, CP and trauma.	Despite efficiency of PP resections, relapse common Increased susceptibility to exocrine and endocrine dysfunction High risk of developing T3cDM or worsening metabolic status when diagnosed with DM prior to surgical intervention. DP results in increased incidence of postoperative T3cDM
Kausch-Whipple (also called the Whipple or PD)		
b. Duodenal-sparing pancreatic head resection		
c. Modified Puestow (longitudinal PJ)		
d. Beger (resection of the pancreatic head with end-to-side PJ)		
e. Frey (coring out of the pancreatic head with longitudinal PJ)		
f. Central pancreatectomy		
g. DP		
2. **TP**	Obligatory surgical procedure for locally advanced or centrally located pancreatic neoplasms, trauma, CP or ARP. Modern treatment regimens, interdisciplinary management and enhanced post-operative care in TP subjects have resulted in long-term survival, better QoL, diminished pain and diminished rates of mortality and morbidity.	TP results in complete endocrine and exocrine insufficiency often resulting with severe metabolic outcomes such as steatohepatitis, malabsorption, difficult glycemic control (brittle diabetes) and liver failure.
3. **TPIAT**	An established dual procedure for a subset of patients with CP and ARP Mitigates post TP-induced T3cDM as much as possible, along with all other advantages that come with performing the TP procedure	Recurrence of DM/brittle diabetes, recurrence of pain, exocrine insufficiency and altered intestinal anatomy and dysfunction.

Abbreviations: ARP—acute recurring pancreatitis; CP—chronic pancreatitis; DM—diabetes mellitus; DP—distal pancreatectomy; PD—pancreaticoduodenectomy; PJ—pancreatojejunostomy; PP—partial pancreatic; QoL—quality of life; TP—total pancreatectomy; TPIAT—total pancreatectomy with islet autotransplantation.

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
