# Peer review of "Challenges of Managing Type 3c Diabetes in the Context of Pancreatic Resection, Cancer and Trauma"

_jcm, 2024, doi:10.3390/jcm13102993_

Round 1
Reviewer 1 Report
Comments and Suggestions for Authors
The following comments may help improve the manuscript:
1. The term “Type 3c Diabetes” that the authors used in the title and throughout the manuscript is outdated and wrong for the reasons outlined in Eur J Endocrinol. 2021 Apr;184(4):R151-R163. Instead, the term “Diabetes of the Exocrine Pancreas” is advocated.
2. The same guidelines also clearly explains why diabetes after pancreatic surgery is NOT part of Diabetes of the Exocrine Pancreas. It is a post-operative complication and should be positioned as such throughout the manuscript.
3. Pancreatic cancer is a legit part of Diabetes of the Exocrine Pancreas.
4. The introduction and management sections contains limited information. Further the management section talks about diabetes secondary to pancreatitis, which appears NOT to be the focus of the present paper.
Comments on the Quality of English LanguageEnglish may need to be reviewed for style and punctuation.
Reviewer 2 Report
Comments and Suggestions for Authors
In detailed review of the subject by the authors. This review will surely help the clinicians to screen the patients appropriately for early identification of diabetes, early management and prevention of diabetes complications. Also it may also help in identifying pancreatic malignancy in suspected patients and early diagnosis might be fruitful.
There are 3 comments from my side.
1. There is an Indian study which has shown that OGTT when performed in CP patients helps in early identification of pre diabetes and diabetes and this if appropriately managed will help in preventing the progression of disease.
2. Also another Indian study has shown the risk factors for early development of diabetes in pancreatic patients. So aggressive screening in such patients can help in early identification of diabetes in such patients.
3. Bach2 gene if positive again is associated with advanced disease and may develop diabetes.
Kindly look into those points.
Reviewer 3 Report
Comments and Suggestions for Authors
This is a very interesting REVIEW article, I personally think it is well written. But I still have some suggestions for the author: 1. I suggest that the sub-title should be consistent with the classification of FIGURE 2. For example, in FIGURE, it says pancreatic tumors, but in the sub-title category, it says pancreatic cancer. 2. I think it would be more understandable if a table could be added to the paragraphs to compare the advantages and disadvantages of different surgical methods or treatment methods. All in all, this is a well-organized literature review, and I think it will also be helpful for clinical treatment.
